# Intention Estimation via Gaze for Robot Guidance in Hierarchical Tasks

**Yifan Shen**                                    yshenaz@connect.ust.hk
**Xiaoyu Mo**                                     xmoac@connect.ust.hk
*Hong Kong University of Science and Technology*

**Vytas Krisciunas**                              vytas@hansonrobotics.com
**David Hanson**                                  david@hansonrobotics.com
*Hanson Robotics Ltd*

**Bertram Shi**                                   eebert@ust.hk
*Hong Kong University of Science and Technology*

**Editor:** Editor's name

## Abstract

To provide effective guidance to a human agent performing hierarchical tasks, a robot must determine the level at which to provide guidance. This relies on estimating the agent's intention at each level of the hierarchy. Unfortunately, observations of task-related movements only provide direct information about intention at the lowest level. In addition, lower level tasks may be shared. The resulting ambiguity impairs timely estimation of higher level intent. This can be resolved by incorporating observations of secondary behaviors like gaze. We propose a probabilistic framework enabling robot guidance in hierarchical tasks via intention estimation from observations of both task-related movements and eye gaze. Experiments with a virtual humanoid robot demonstrate that gaze is a very powerful cue that largely compensates for simplifying assumptions made in modelling task-related movements, enabling a robot controlled by our framework to nearly match the performance of a human wizard. We examine the effect of gaze in improving both the precision and timeliness of guidance cue generation, finding that while both improve with gaze, improvements in timeliness are more significant. Our results suggest that gaze observations are critical in achieving natural and fluid human-robot collaboration, which may enable human agents to undertake significantly more complex tasks and perform them more safely and effectively, than possible without guidance.

**Keywords:** Gaze, Intention Estimation, Hierarchical Task, Robot Guidance, Humanoid Robot

## 1. Introduction

Humans can offer effective guidance in different roles, such as mentor (Marcdante and Simpson, 2018), teacher (Stokhof et al., 2017), coach (Lewis et al.), etc. They rely on not only spoken language and observation of directly task-related movements, but also on observations of secondary behaviours, such as eye gaze (Kredel et al., 2017; Brennan et al., 2008; Schneider and Pea, 2017; Wang et al., 2019). Gaze is an important cue that helps people to estimate others' intention (Foulsham, 2014; Newn et al., 2017; Neider et al., 2010) and confidence (Emhardt et al., 2020), which is often crucial for proper guidance.

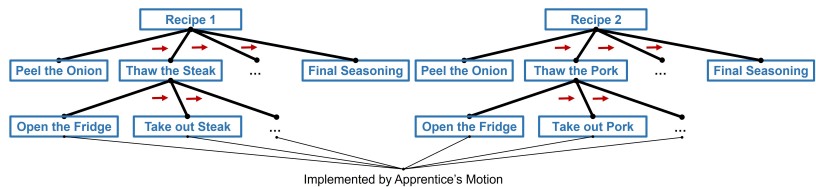

Figure 1: Hierarchical tasks in a kitchen scenario represented as a tree structure. The highest level of the hierarchy (root node) corresponds to the recipe being followed. Leaf nodes result in observable movements by the actor. Arrows indicate the order in which tasks must be executed.

Here, we propose a framework that enables a humanoid robot to provide guidance to a human performing a hierarchical task based on observations of the human's task-related movements and intent estimation from eye gaze. While past work on human-robot collaboration has studied robot guidance in tasks such as coaching (Fasola and Matarić, 2013), teaching (Conati et al., 1997) or navigating (Kanda et al., 2009), to the best of our knowledge, eye gaze has not been exploited for robot guidance. More work has been done on intention estimation from eye gaze, which has applied to applications ranging from eye typing (Pi and Shi, 2017) to control of wheel chairs (Cojocaru et al., 2019), robotic arms (Li et al., 2017) or exoskeletons (Frisoli et al., 2012). (Huang et al., 2019) studied the inverse problem of a robot student generating gaze towards human teachers. Gaze has also been exploited in action recognition (Li et al.). However, in these studies tasks typically have a flat structure. Intention estimation finds the most probable goal from among a set of similar alternatives, e.g. desired destinations or directions of movement. Little attention has been paid to intention estimation from eye gaze for tasks with a multi-level hierarchical structure (Erol et al., 1996; Conati et al., 1997; Erol et al., 1994).

One of the main problems in providing guidance in hierarchical tasks is to determine the level at which guidance is needed. For example, consider a kitchen scenario where an apprentice is making a meal under the guidance of a master chef (Figure 1). The apprentice might encounter difficulty at different levels of the task hierarchy, requiring different cues from the master chef. For example, suppose that the apprentice has peeled the onion and opened the refrigerator, then stops. This might be due to confusion at a high level (the apprentice is unclear which recipe they are following) or at a lower level (the apprentice knows to follow Recipe 1, but cannot find the steak). To provide precise guidance, the master chef must estimate the apprentice's intention, i.e., which tasks are being executed at each level of the hierarchy.

Intention estimation at all levels of the hierarchy based only on observations of the apprentice's task-related movements is difficult. Instantaneous observations typically provide direct information about task performance at the lowest level of the hierarchy, but different high level tasks might contain the same lower level tasks. For example, the task "peeling the onion" is consistent with both Recipes 1 and 2. Thus, based on this observation alone, the master chef cannot determine whether the apprentice is following the correct recipe, and must wait until the apprentice reaches for the steak or the pork to disambiguate this. This ambiguity impairs the timeliness of the cues being delivered.

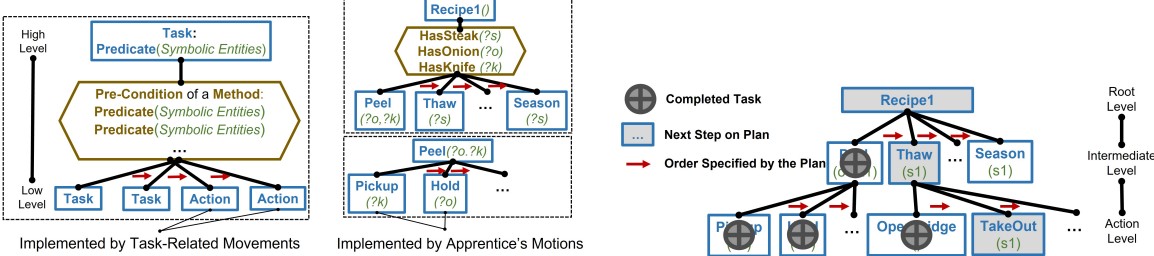

(a) The relation between task, action and method in an HTN.

(b) A specific example of an HTN from the kitchen scenario.

(c) Illustration of an agent progressing through the plan for task "Recipe1".

Figure 2: Planning with HTN.

These ambiguities leading to imprecise and poorly timed cues can be resolved by observations of secondary behaviors, such as gaze and facial expressions. In the first example above, the master chef might have high confidence that the apprentice knows which recipe to follow since he saw the apprentice looking at the recipe earlier, before the apprentice started any cooking related movements. In the second example, the chef might provide guidance earlier if the apprentice looks at the master chef and/or expresses confusion through facial expression.

This work proposes a framework for integrating observations of task-related movements and gaze, enabling a humanoid robot to provide timely and precise guidance to a human agent performing hierarchical tasks. We focus on humanoid robots, since the human tendency to anthropomorphize (Damiano and Dumouchel, 2018)(Fussell et al., 2008) will evoke social interactions (Damiano and Dumouchel, 2018) and elicit corresponding gaze behaviours (Mukawa et al., 2001), which can be exploited for disambiguation. We implemented and tested the framework in virtual reality and compared its performance against that of a human wizard to validate its effectiveness and to understand the contribution of gaze.

## 2. Representation and Analysis of Hierarchical Tasks

The Hierarchical Task Network (HTN) is a classic representation of hierarchical tasks for planning (Erol et al., 1994). An HTN represents the state of the world as a set of atomic predicates. The domain is described by a set of *task*s, *method*s and *action*s (Figure 2). A task represents an abstract goal, e.g., following a recipe. Methods describe different ways to complete a task by decomposing it into a set of (sub-)tasks and / or actions. Methods have a tree structure and organize tasks into different levels of abstraction. Methods are applicable when a set of preconditions are met. The same sub-task or action could be involved in several different task and in different decomposition of the same task. An action is a special task which cannot be further decomposed and is implemented physically through task-related movements, resulting in changes of the world state. Tasks, actions and preconditions of methods are represented as predicates acting upon symbolic arguments. Physical entities involved in a task can be explicitly represented by these symbolic arguments or implicitly bound to the predicate, e.g., fridge in "OpenFridge". To find a plan to achieve a certain task,

an HTN planner recursively chooses and applies applicable methods to find a decomposition of the goal task. A plan is found when the process terminates with a sequence of actions.

Figure 2(c) shows an agent progressing through a plan for the task "Recipe 1." The plan inherits a tree structure from the methods, where the root node represents the overarching goal, lower level vertices on the tree represent intermediate sub-goals, and leaf nodes correspond to actions. Although the tasks in a decomposition might be partially ordered, a plan contains totally-ordered sequences of tasks at each level, as shown by the arrows in the figure. At each step of the plan, an agent is simultaneously executing tasks at different levels (e.g. actions at the leaves and intermediate sub-goals at higher levels of the tree). Thus, the current steps along the plan corresponds to one branch of the tree, shown as grey boxes in the figure.

Under this formulation, an agent carrying out a plan could have difficulty with tasks at different levels of the current branch. Guidance should be provided according to the agent's intention, which we define as as a set of binary variables: one for each level indicating that the agent is either aware of, executing, or intending to execute the corresponding task. Ideally, guidance should be based on the agent's actual intention. For example, if the binary variable for the root node is 0, the agent might need reminding about which recipe they should be following. If the binary value for the root node is 1, but 0 for a lower level node, the agent might need a reminder about what the next step in the recipe should be. In practice, guidance is based on estimates of the agent's intention, since the intention is not directly observable from actions, except at the lowest level of the hierarchy and only if the agent is actually executing the action. As described above, this leads to ambiguities. These can be resolved, at least partially, by exploiting gaze, as we describe below.

Planning techniques have been adopted previously to estimate human intention. This is often referred to as *plan / goal / intention recognition as planning* (Sohrabi et al., 2016) (Meneguzzi and Fraga Pereira, 2021). Recently, Singh et al. estimated a player's intention in a turn-based game using planning (Singh et al., 2020). They assumed the likelihood of past actions given a potential goal to be proportional to the similarity between the global optimal plan for that goal and the optimal plan for that goal given the action history as a prefix. They also showed that incorporating gaze improves accuracy and reduces computation cost by pruning expensive paths. Whereas they used planning in a non-hierarchical task to assign probabilities to different potential goals, we use planning to find the optimal current step in executing a single hierarchical task with a fixed goal. We use gaze to estimate intention at different levels of the hierarchy, whereas they use gaze to estimate intention to achieve different goals.

## 3. Proposed Framework

Our proposed guidance framework consists of three components: an HTN planner, an intention estimator and a guidance controller (Figure 3(a)). The HTN planner receives observations about the task-related movements of the agent and maintains the world state of the HTN domain. Given the root task and the world state, the planner finds the set of tasks the agent is currently expected to execute, which we refer to as the *optimal branch*. The intention estimator then estimates the agent's intention at each level of the optimal branch. These estimates are conditional probabilities given observations of the agent's movements

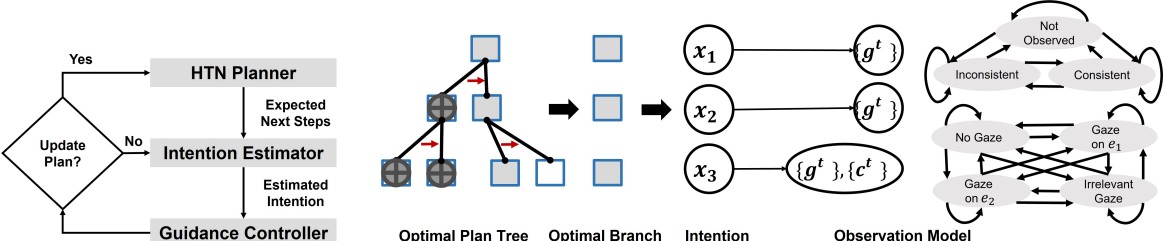

($a$) Flowchart of the framework.

($b$) Steps followed by the intention estimator.

Figure 3: Overview of the framework.

and gaze. The guidance controller generates verbal and gestural cues to the agent based on the estimated intention. Intuitively, if all probabilities are high, the controller gives no cue. If one or more of the probabilities is low, the controller gives a cue appropriate for the highest level at which the probability is low.

### 3.1. HTN Planner

The planner keeps track of actions, which are executed via observable task-related movements, and maintains the world state for planning. We make the simplifying assumption that the agent is perfectly rational, i.e., follows the optimal plan. We use HATP (Lallement et al., 2014) (released under 2-clause BSD license) to find the optimal plan and extract the optimal branch, corresponding to what the agent should be doing at the current time. However, any other HTN planner such as SHOP2 (Nau et al., 2003) could be used instead.

### 3.2. Intention Estimation

Figure 3($b$) illustrates the steps followed for estimating intention at the current time step $T$. Let $\{A_i, i = 1, 2, ..., N\}$ denote tasks on the branch, where $N$ is the number of levels in the branch, $A_1$ is the goal and $A_N$ the leaf action. The intention is a set of independent, binary-valued random variables $\{x_i, i = 1, 2, ..., N\}$, where $x_i = 1$ if the agent is executing, intending to execute, or at least aware of the need to execute $A_i$. To estimate intention, we compute the probability that $\{x_i = 1\}$ given the past history of task-related movements $c^t$ and gaze $g^t$ up to time step $T$. By Bayes rule

$$P(x_i | \{c^t\}_{t=1}^T, \{g^t\}_{t=1}^T) \propto P(\{c^t\}_{t=1}^T, \{g^t\}_{t=1}^T | x_i) \tag{1}$$

$$\propto P(\{g^t\}_{t=1}^T | x_i) P(\{c^t\}_{t=1}^T | x_i) \tag{2}$$

where in (2) we assume movement and gaze are conditionally independent given the intention variable $x_i$. The conditional probabilities for different levels $i$ are estimated separately, based on the same observed movements and gaze history. We construct $N$ gaze models, one per level of the hierarchy $x_i$. However, we compute the conditional probability of the the movement sequence only at the lowest level, $P(\{c^t\}_{t=1}^T | x_N)$, corresponding to actions. This assumes the movement sequence to be independent of the intermediate goals, which is certainly not true, but given the ambiguity pointed out above, is a reasonable simplifying

assumption. More realistic movement models that do not assume independence could be easily incorporated. We discuss this issue further in Section 6.

We treat $c^t$ as a discrete random variable that assumes one of three values: "not observed", "consistent" or "inconsistent", where the consistency is defined with respect to action $A_N$. For instance, the apprentice's arm reaching for the fridge door is consistent with the action "OpenFridge". We model $P(\{c^t\}_{t=1}^T | x_N)$ with two 3-state Markov Chains: one for $x_N = 1$ and one for $x_N = 0$. Intuitively, sequences with many movements consistent with $A_N$ are more likely if $x_N = 1$ than if $x_N = 0$.

We also treat $g^t$ as a discrete random variable. The number of possible values depends upon the number of physical entities associated with the task $A_i$, which we can extract automatically from the HTN based on the predicate describing $A_i$ and the method for $A_i$ used in the plan. Assuming a total of $M_i$ entities $\{e_1, e_2, ..., e_{M_i}\}$ involved in task $A_i$, $g_t$ assumes one of $M_i + 2$ values: $M_i$ values indicating the gaze falls on one of the associated entities $e_{m_i}$, one value indicating the gaze does not fall on any of the relevant entities, and one value indicating no gaze observation. We model $P(\{g^t\}_{t=1}^T | x_i)$ with two $(M_i + 2)$-state Markov Chains: one for $x_N = 1$ and one for $x_N = 0$. Distinguishing between gaze towards different entities enables us to model task-specific gaze patterns, e.g., alternations of the apprentice's gaze between the onion entity and the knife entity when performing the task "Peel(onion,knife)" in Figure 2(b).

At this point it is worth recapping the kitchen example. If the apprentice has a wrong recipe in mind, s/he is likely to look at the wrong one, which is an entity not present in the predicate of task "Recipe1()" (Figure 2(c)). Therefore, gaze on the wrong recipe is inconsistent, causing $P(\{g^t\}_{t=1}^T | x_{\text{Recipe1}})$ to drop for $x_{\text{Recipe1}} = 1$. On the other hand, if everything is fine, gaze on the steak is consistent with the tasks "Recipe1()", "Thaw(steak)" and "TakeOut(steak)" at all levels of the hierarchy in Figure 2(c), since the entity $e_{\text{steak}}$ is relevant to all of those tasks. Intuitively, entities relevant to tasks on the optimal branch are organized hierarchically. High-level tasks have more related entities, whereas lower-level tasks have fewer related entities: a subset of those in the higher-level task. Thus, gaze models at higher levels are more spatially diffuse than gaze models at lower levels, which are more spatially localized. These differences in modelled gaze behavior enable the framework to use gaze for disambiguation among different levels of the pyramid.

We add an additional "not seeking guidance" variable $x_0$, which indicates whether the agent is seeking guidance ($x_0 = 0$) or not ($x_0 = 1$). In our example, when needing help, the apprentice might look towards the master chef. Our framework assumes a humanoid robot guide, which many people tend to anthropomorphize (Damiano and Dumouchel, 2018)(Fussell et al., 2008). The associated gaze feature assumes three values: "no-observation", "looking at the robot" and "looking elsewhere". Associated gaze trajectories are modelled by 3-state Markov chains, similar to the above.

## 3.3. Guidance Controller

The guidance controller maps the estimated intention, $\{P(x_i|\cdot), i = 0, 1, 2, ..., N\}$ where $\cdot$ is an abbreviation for the movement/gaze trajectory, to a cue depending upon whether the agent *wants* guidance ($P(x_0 = 1|\cdot)$ is low), or whether the agent *needs* guidance ($P(x_i = 1|\cdot)$ is low for some $i$ between 1 and $N$). The guidance controller first evaluates whether the

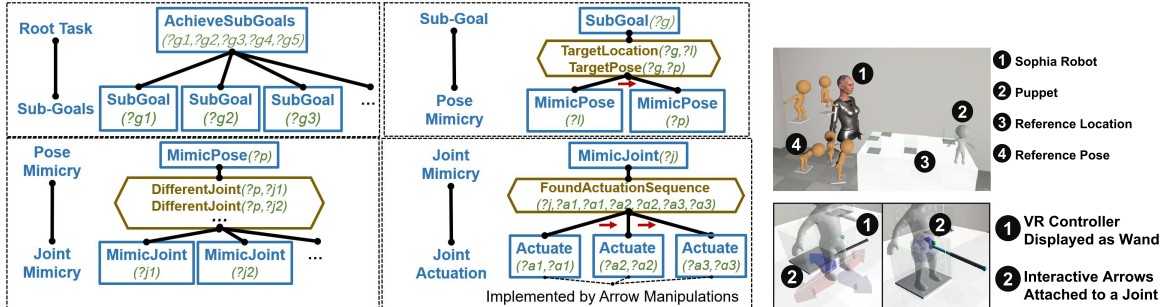

(a) HTN representation of the task.

(b) A demonstration of the task environment in VR and the arrow manipulations.

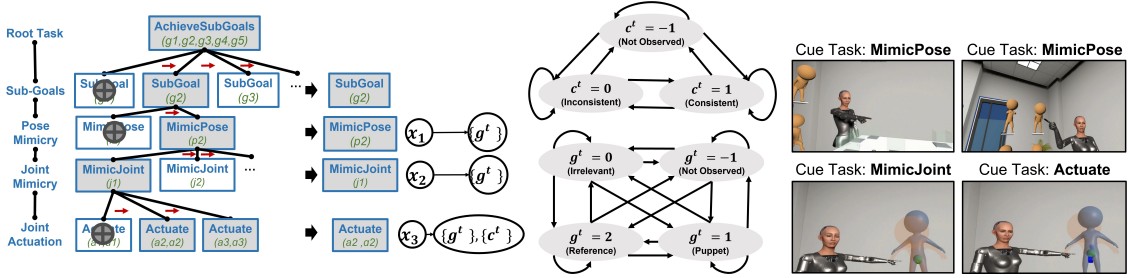

(c) A summary of the implementation of the framework.

(d) An illustration of cues.

Figure 4: Overview of the VR posing task and the implementation of the framework.

agent needs guidance. Formally, given a threshold $h$,

$$\text{Provide guidance if} \quad \exists i: \ P(x_i = 1|\cdot) \leq h, i \in \{1, 2, ..., N\} \tag{3}$$

$$\text{Choose cue at level } i \text{ where } i = \max\{n | P(x_n = 1|\cdot) \leq h, n = 1, 2, ..., N\} \tag{4}$$

If the agent does not need guidance, the controller checks whether s/he *wants* guidance. Formally,

$$\text{Provide guidance if} \quad P(x_0 = 1|\cdot) \leq h \tag{5}$$

$$\text{Choose cue at level } i \text{ where } i = \text{argmin}_n\{P(x_n = 1|\cdot), n = 1, 2, ..., N\} \tag{6}$$

The guidance cues are both task and level dependent. If guidance is needed and there are multiple levels of confusion, the highest level needed cue is given. If guidance is wanted, the controller provides the cue at the level with the lowest estimated intention.

## 4. Experiment

We designed experiments in virtual reality (VR) to test two hypotheses. First, a robot equipped with the proposed framework, acts as an effective helper and offers timely and precise guidance. Second, the performance of the proposed framework benefits significantly by incorporating gaze information.

### 4.1. VR Posing Task

The proposed framework was implemented in a VR posing task. Human agents were asked to move a puppet through a sequence of spatial locations, where the puppet should assume a different pose at each location. The task is motivated by an on-going collaboration with a local hospital seeking to use humanoid robots to guide patients through various check-in procedures, such as an entrance interview and measurement of vital signs. Figure 4(b) shows the task environment, which consists of a puppet, a set of reference objects indicating locations and poses, and a virtual Sophia robot. The posing task is carried out by the human subject. Sophia serves as a guide, providing a description of the task at the start of each trial, and cues in the form of spoken utterances and body gestures (e.g. pointing) while the agent is performing the task. In Figure 4(a) the posing task is presented as an HTN. To complete the *root task*, the agent needs to achieve five *sub-goals*. Two tasks need to be completed for each sub-goal: translating the puppet to a desired location and controlling the puppet to mimic a reference pose. For ease of description, we treat both as *pose mimicry* tasks. A pose mimicry task in turn requires the completion of a set of *joint mimicry* tasks. Joint mimicry is done by executing a sequence of *joint actuation task*s around different axes by different angles. The task "Actuate" at the joint actuation level is the action of this HTN which could be implemented by the task-related movement of *arrow manipulation*. A set of interactive arrows similar to those in (Leeper et al., 2012) is attached to each joint and could be manipulated by the agent via the VR controller (Figure 4(b)). Every button press of the controller causes the selected arrow to actuate the joint around an axis by a fixed amount, hence an "Actuate" action requires repeated manipulations of the correct arrow. Figure 4(d) shows the cue Sophia might offer on each non-trivial task. For a more lively view of cues check this demo video. A demo of a human executing the posing task could be found here. The VR posing task is implemented in Unity on a free personal license. The Unity-based assets we used are listed in Appendix A.3.

Figure 4(c) summarizes the implementation of the proposed framework for the VR posing task. A task-specific planner for the posing task is created (Appendix A.1), consisting of a symbolic planner (for arranging tasks at sub-goal and pose mimicry levels) and a motion planner (for arranging tasks at joint mimicry and joint actuation levels). For estimating intention, we create the intention variables at three levels: the pose mimicry, joint mimicry and joint actuation levels (from highest to lowest level of the hierarchy). The relevant entities for the pose mimicry tasks are the puppet and the reference object. For lower-level joint mimicry (or joint actuation) tasks, the related entities are specific joints (or arrows) on the puppet and the reference object. See Appendix A.2 for more details. As discussed above, related entities at lower levels are subsets of the related entities at higher levels, leading to more spatially localized gaze models at lower levels. The guidance controller chooses from the set of verbal/gestural cues shown in Figure 4(d).

### 4.2. Experimental Protocol

A total of 21 subjects were recruited and partitioned into two groups: the *Automated Group* (7 male and 4 female) and the *Wizard Group* (5 male and 5 female). Subjects in both groups completed two task sessions. For the Automated Group, cues were generated using the proposed framework. In one session our framework had access to both gaze and task-related

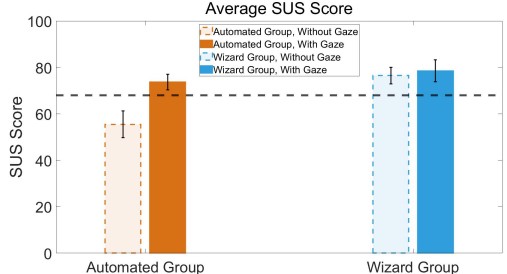 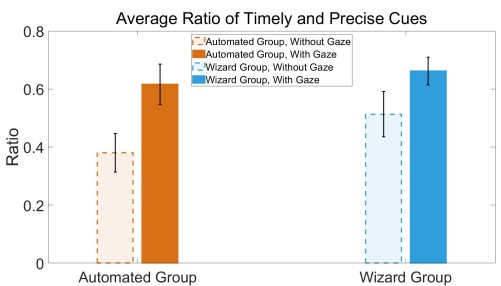

(a) Average SUS score where the dashed line is placed at 68, above which a system is usable (Brooke, 1996).

(b) Average ratio of timely and precise cues. The error bars indicate standard error across sessions.

Figure 5: Measurements averaged over all sessions.

movements (arrow manipulation commands), whereas in the other gaze was not observable. The same parameters were used for all subjects across all sessions. For the Wizard Group, a human wizard, who had access to the same set of cues available to our framework, chose the cues. The wizard observed the task progress through a monitor, which also showed both the agent's arrow commands and eye gaze in one session and only the arrow commands in the other session. See Appendix B.6 for more details. We followed the same procedures for both groups and regardless of whether gaze is used. Subjects in the Wizard Group were unaware of the wizard. We spaced the two sessions by several days to control skill build-up and randomized the order of with-gaze and without-gaze sessions. Appendix B.3 gives more details on the experimental protocol, which was approved by the Human and Artefacts Research Ethics Committee at the Hong Kong University of Science and Technology as HREP-2021-0193.

### 4.3. Evaluation Criteria

To measure the overall usability of Sophia as an helper, the subject is asked to complete the questionnaire of System Usability Scale (Brooke, 1996) (Appendix B.5) after completing the posing task. We used interviews to evaluate the timeliness and precision of the guidance provided. After completing a posing task, the subject would watch a video playback of the session. At the end of each cue provided by Sophia, the subject answered three multiple-choice questions (Appendix B.4). The first question asked whether there *should* be a cue. If yes, the second and third questions asked about the timeliness and precision (level) of the cue. When answering the question regarding timeliness, the subject was instructed to ignore the actual content of the cue, and focus on whether a cue was needed at the time it was issued.

### 5. Results

Figure 5 illustrates the SUS score and the ratio of cues labeled both timely and precise within a session averaged across all sessions. When gaze is available our framework achieves performance similar as the Wizard Group in these criteria, supporting Hypothesis 1. The

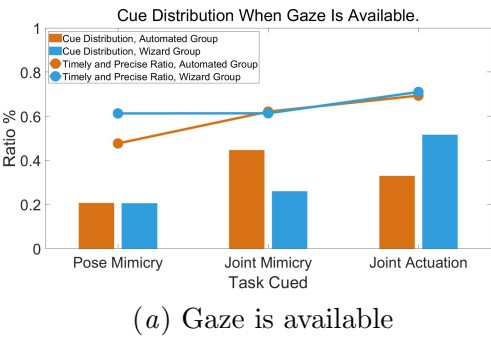 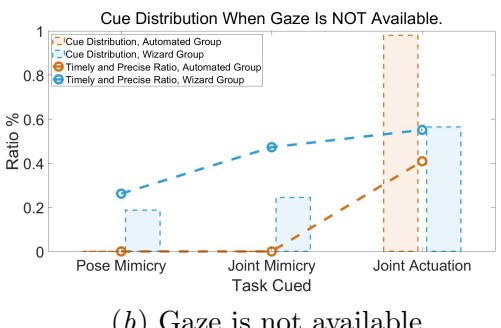

(a) Gaze is available  (b) Gaze is not available

Figure 6: Distribution of cues computed by dividing the number of cues of each level by the total number of cues from all sessions. The ratio of timely and precise cues at each level are shown by the circles.

bar plot representing cue distributions computed across sessions in Figure 6(a) indicates that our framework generates diverse cues across different levels like the human wizard, and that it does so with good timing and precision, as reflected by the circles representing the ratio of timely and precise cues at each level.

To validate Hypothesis 2 we first make a within-group comparison for the Automated Group. Figure 5 shows that without gaze, the framework is unusable (SUS below 68) and there is a significant drop in the quality of guidance. The cause is two-fold. Without gaze the framework only provides guidance at the action-level $A_N$ (Section 3.2), leading to the skewed distribution of cues in Figure 6(b). Comparing the filled and empty circles for the "Joint Actuation" level in Figures 7(c) and 7(d) shows that, without gaze, the timing of guidance degraded more dramatically than the precision of guidance. Thus, poor cue timing is the primary reason for the degradation without gaze. Intuitively, this is expected. If only observations of arrow manipulation are available, the framework cannot tell if the subject is off track until s/he makes drastically wrong movements, leading to cues that come too late. One example is given in this demo.

Qualitatively similar, but smaller, performance drops are seen in the Wizard Group. Although usability is maintained (Figure 5(a)), the quality of guidance drops significantly (Figure 5(b)). Similar to the results with our framework, blue circles in Figure 7 reveal the timeliness of guidance to be the limiting factor of performance. This is especially clear for cues at the pose mimicry task level, which is not surprising. Since the pose mimicry task sits at the highest level of the hierarchy, it is the most subject to ambiguity given observations of arrow manipulations. Indeed, the wizard remarked that s/he had to adopt a strategy similar to our framework when gaze information was absent, waiting until the subject made clearly incorrect or unexpected arrow manipulations before providing a cue at the pose mimicry level. This video shows one such occasion from a Wizard Group session. This anecdotal evidence further supports Hypothesis 2.

We also make a between-group comparison when gaze was not available. Figure 6(b) shows that unlike our framework, the human wizard still provides cues at different levels in the absence of gaze. Figure 7 shows that the wizard provides higher quality cues than our

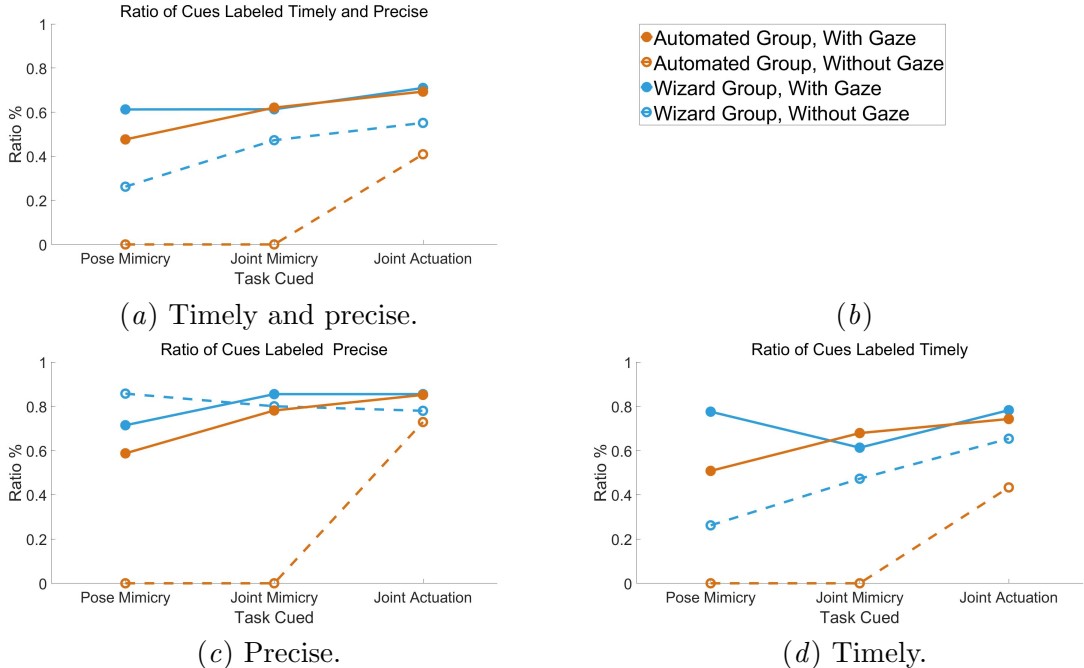

(a) Timely and precise.

(b)

(c) Precise.

(d) Timely.

Figure 7: Ratio of timely and / or precise cues per type.

framework, especially in terms of timeliness. This is expected given the simplified movement models $P(\{c^t\}_{t=1}^T | x_i)$.

## 6. Discussion

Our framework assumes intention variables at each level are defined only for the optimal branch, even though alternative plans leading to the same goal exist. While our framework can adapt the changes in the plan, re-planning does not occur until the person has finished executing a task. If the human agent follows a sub-optimal (but equally viable) plan, our framework might provide unwanted guidance. This assumption also leaves little room for creativity, e.g., if the agent wishes to exclude some part of the recipe based on preference.

Our framework also assumes that intention variables at different levels are independent, and that motion $\{c^t\}_{t=1}^T$ and intent $x_i$ are independent for non-leaf tasks. The effects of this over-simplification are especially clear when gaze is not available. The distributions of cues generated by our framework and by the human wizard are quite different (Figure 6(b)). Unlike the human wizard, our framework generates no cues at the higher levels of the hierarchy.

To address these shortcomings, we could (and should) use a more complex inference model that incorporates more information from within and outside the optimal branch. Much information about high level intent can be extracted from observed movements by reasoning through the tree. For example, the subject performing a motion uniquely associated with a leaf node task that only appears in one higher level action provides strong evidence that the subject intends to perform the high level task.

Despite their shortcomings, these dramatic oversimplifications serve positive purpose in highlighting the power of the gaze cue. When gaze is available, our framework generates cues nearly as well as the human wizard (Figure 5 and Figure 6(a)). This suggests that there may be little gained by additional observation of task-related behavior. This is consistent with impressions of the human wizard, who remarked that providing guidance was difficult without gaze because deducing higher level intent became more difficult. In sessions where gaze is available, the wizard relied heavily on the subject's gaze patterns to estimate high-level intention, as reflected in this video. Interestingly, in the above example the wizard identifies the subject's wrong intention at the pose-mimicry level in a way close to what our framework does in a similar situation. The efficacy of gaze allows our framework to compensate for its flawed movement model and to achieve human-level performance.

The HTN facilitates automatic gaze model construction, as we can identify the entities involved in each task from its task and method definitions. Although this specifies the spatial locations likely to be gazed at during task performance, it does not include information about the temporal components of the gaze trajectory, which we modelled using Markov Chains. In this work, we hand-tuned the transition parameters to encode our empirical finding that the agent looks at irrelevant entities more frequently when executing the wrong task. Future work should address automated construction of the gaze models, as well as the extension to continuous, rather than discrete gaze distributions.

Some problems relating to cue timeliness and precision might be addressed by incorporating some form of Theory-of-Mind into the framework to model agent state and the effect of past history (Devin and Alami, 2016; Favier et al., 2022). Figure 7(d) indicates that the timeliness of guidance from our framework is comparable to that from the wizard for lower level tasks, but worse at the highest level (pose mimicry). We suspect that the main reason is that our framework cannot adapt to the agent's pace of execution, whereas the human wizard can modify his/her rate of feedback based on past observations of the agent's behavior. Our framework also generates unwanted cues most frequently at the pose-mimicry level. We suspect this is because the conditional probabilities $p(x_i|\cdot)$ are reset to the same initial values after each cue. A human agent typically retains high-level intent for a long time after receiving a high-level cue, but our framework retains no memory of past cues, possibly resulting in repeated cue generation. The reset also means that if the agent misses a cue, s/he must wait for the probabilities to build up again.

In addition to the directions pointed out above, our model could be improved in many other directions. The posing task we considered has only one type of task per level. Future work should examine plan trees of varying depth with more task types per level. Humans provide different forms of guidance for the same task. The guidance controller could be extended to provide increased granularity of cues. We collected only subjective assessments. Future studies could collect objective metrics, such as task completion time and joint movement trajectories. Further work could be done to distinguish between the *need* versus the *desire* for guidance. Our framework partially incorporates this with the "not seeking guidance" variable and partially assessed it in the the first multiple-choice question (Appendix B.4), but much work remains to be done.

## 7. Conclusion

We have proposed a framework for robot guidance to a human agent performing hierarchical tasks that incorporates observations of task-related movements and gaze to estimate the agent's intention. We implemented and evaluated the framework in a VR posing task. Comparisons with a human wizard reveals that our framework provides guidance close to human-level performance, in terms of overall usability and timeliness and precision of guidance. Ablation experiments where gaze was removed shows that the high level of performance is a direct result of incorporating gaze information to resolve ambiguities inherent in the hierarchical structure of the task.

## Acknowledgments

This work was supported in part by the Hong Kong Research Grants Council under Grant 16214821 and in part by seed funding from the Hong Kong University of Science and Technology's Center for Aging Science.

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

## Appendix A. Further Details Regarding the Implementation of the Proposed Framework in the Posing Task

### A.1. Planner

**Task and Motion Planners**   As illustrated in Figure Figure 4(a), the HTN representing the posing task consists of both symbolic tasks (sub-goals and pose mimicry tasks) that has no direct kinematic meanings as well as motion tasks (joint mimicry tasks and joint actuation tasks) that could be mapped to re-configurations of the puppet. Therefore the planner needs both symbolic task planning and motion planning. For symbolic planning, as stated in Section 3.1, we use the HATP planner proposed in (Lallement et al., 2014) and released under 2-clause BSD license to solve the symbolic part of the posing task given its progress so far, where the optimality is measured by the total amount of rotation and translation in the puppet necessitated by a plan. The optimal branch of the partial plan found by HATP would then be sent to the motion planner since it still lacks joint mimicry and joint actuation tasks the person is expected to execute. By default, at joint mimicry level the planner would choose the most-different joint on the puppet with respect to the target pose as the next step. At joint actuation level, if the joint of interest is prismatic the plan could be trivially found. On the other hand, when planning for a revolute joint the planner decides which arrow the subject would use first by exhausting all possible conventions of Euler angle decomposition with respect to the rotation transform needed at the joint and choose the first step of the Euler angle sequence with minimal total rotation. With tasks at these two levels appended, the framework now has the complete optimal branch consisting of expected tasks the subject would be performing at all levels of the hierarchy.

**Re-plan**   The re-plan behaviour of the planner when the world state gets updated is also two-fold. For the symbolic planner, if the subject completes a pose mimicry task like moving the puppet to a location – expected or unexpected – the planner re-plans by first invoking HATP and then the motion planner. In contrast, should the subject finish mimicking a joint or choose to first adjust a joint other than the expected one, the planner re-plans by sending the same symbolic plan to the motion planner again. Similarly, if the subject veers off the expected joint actuation sequence the planner re-computes a new sequence given the expected joint mimicry task and the current joint configuration.

**Think**   To model the think process of subjects upon task completion, a think-phase is added whenever the subject completes a task, during which the intention estimation is skipped.

### A.2. Intention Estimation

**Discretized Features**   In the posing task we set

$$c^t = \begin{cases} -1 & \text{if no arrow manipulation is observed} \\ 0 & \text{if the arrow manipulation is inconsistent w.r.t. joint actuation task} A_N \\ 1 & \text{if the arrow manipulation is consistent w.r.t. joint actuation task} A_N \end{cases} \quad (7)$$

Similarly, for each task $A_i$,

$$
g^t = \begin{cases} -1 & \text{if no gaze is observed} \\ 0 & \text{if looking at irrelevant entities w.r.t. task} A_i \\ 1 & \text{if looking at relevant entities on the puppet w.r.t } A_i \\ 2 & \text{if looking at relevant entities on the reference object w.r.t. task} A_i \end{cases} \tag{8}
$$

where as shown in Figure 8 "relevant entities" could mean the puppet / reference object as a whole, or a joint / arrow on the puppet or the reference object, depending on the task $A_i$ stands for. To determine whether the gaze of the subject falls on one of those entities we use both ray-cast method (Wang et al., 2017) and angular distance threshold.

**Markov Chain Parameter** As shown in Figure $4(c)$ both $P(\{g^t\}_{t=1}^T|x_i)$ and $P(\{c^t\}_{t=1}^T|x_i)$ are modeled by finite-state Markov Chains. We follow empirical rules to select parameters of these Markov Chains. For modeling arrow manipulation with $P(\{c^t\}_{t=1}^T|x_i)$, the subject is more likely to manipulate arrows correctly if executing the right (joint actuation) task, i.e., $P(c^t = 1|x_i = 1, c^{t-1} = \cdot) > P(c^t = 1|x_i = 0, c^{t-1} = \cdot)$. Further, there should at least be some arrow manipulation from the subject when executing the right tasks, hence $P(c^t = -1|x_i = 1, c^{t-1} = -1) < P(c^t = -1|x_i = 0, c^{t-1} = -1)$ which is effectively a timeout if no arrow manipulation has been observed for a long time. For modeling gaze with $P(\{g^t\}_{t=1}^T|x_i)$, it is unlikely for the subject to keep fixating either entities on the puppet, entities on the reference object or irrelevant entities for a prolonged period should he / she is executing the right task, i.e., $P(g^t = g^{t-1}|x_i = 1, g^{t-1}) < P(g^t = g^{t-1}|x_i = 0, g^{t-1})$. Instead, when executing the right task the subject's gaze is expected to alternate between entities on puppet and entities on the reference object, i.e., $P(g^t = 1|x_i = 1, g^{t-1} = 2) > P(g^t = 1|x_i = 0, g^{t-1} = 2)$ and $P(g^t = 2|x_i = 1, g^{t-1} = 1) > P(g^t = 2|x_i = 0, g^{t-1} = 1)$.

### A.3. Unity Assets

Here we list assets used by the implementation of the framework in the posing task with hyper links pointing to their corresponding unity asset store page. All assets are under Standard Unity Asset Store EULA.

- Goodrect Popup

- Ultimate Replay 2.0

- Animancer Lite

- 3D Items - Free Wand Pack

- Hyper-Casual Character Stickman sphere head

- Primitives

- LowPoly Arrows Pack

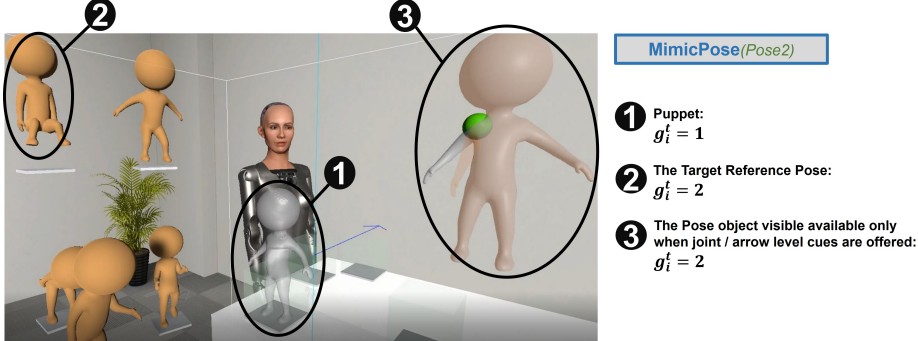

(a) Related entity specification for a pose mimicry task. If the gaze falls on any other entities in the scene the feature value would be set to $g^t = 0$, i.e., irrelevant to the task. If no gaze is captured, $g^t = -1$.

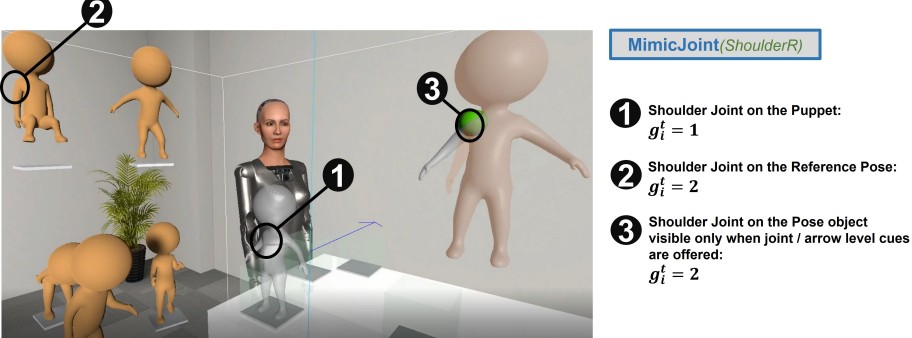

(b) Related entity specification for a joint mimicry task. If the gaze falls on any other entities in the scene the feature value would be set to $g^t = 0$, i.e., irrelevant to the task. If no gaze is captured, $g^t = -1$.

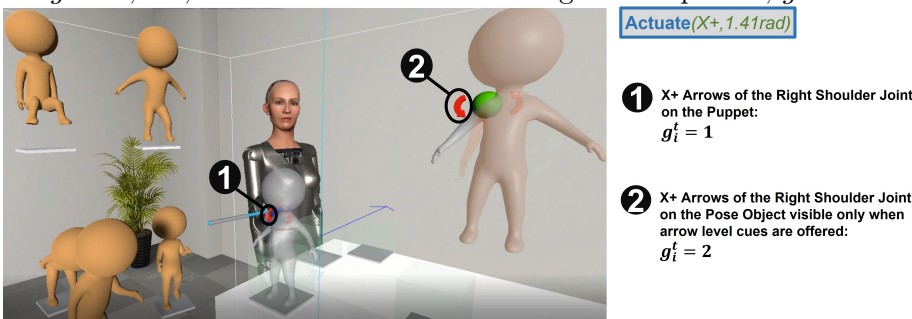

(c) Related entity specification for an joint actuation task. If the gaze falls on any other entities in the scene *when a group of arrow is visible* the feature value would be set to $g^t = 0$, i.e., irrelevant to the task. If no gaze is captured or no arrow was shown, $g^t = -1$. Note that as shown in this demo video the interactive arrows would not appear until the wand has been moved close to the arrows. Also note that for the same direction of rotation of pair of arrows are available, both are treated the same when assigning gaze features.

Figure 8: Related entity specification for assigning gaze feature values, Note that the large pose object in front would not be available until Sophia have offered a joint-level or arrow-level cue, as shown in Figure 4(d)

## Appendix B. Further Details of Experiment

### B.1. Subject and Human Wizard Recruitment

The subjects and the human wizard are all college students but are from different fields, covering engineering, natural science, business and social science. They were informed of the content, estimated duration and payment of the experiment before they give consent and register for the experiment.

### B.2. Curated Data and Privacy

The curated data set contains no personally identifiable information. The replays of the experiment sessions are in VR view only without any photo or video footage involving the subjects. Both the replays and the answers to the SUS questionnaire and the multiple-choice prompts are indexed by subject number alone. The wizard was informed that there would be a follow up interview regarding the opinion and experience formed through Wizard Group sessions and we acquired the wizard's consent to present some of the remarks in an anonymous way in this paper.

### B.3. Experiment Protocol

The same experiment protocol adopted by the two sessions of each of the subjects of the two groups are described here.

1. **Tutorial** A tutorial session would be given to all subjects before the commencement of their first task session, during which the subject is directed to complete a posing task which has the same representation as shown in Figure 4($a$) but consists of simple, mock poses. The purpose of this tutorial session is to familiarize the subject with the task procedures, the arrow manipulation, the role of Sophia and the forms of cues regarding each type of tasks. The mock poses are shown in Figure 9($a$). It would be emphasized at the end of the tutorial session that the subject is supposed to actively explore and progress the task, while Sophia would observe and attempt to provide guidance as seen fit.

2. **Gaze Calibration** The on-board Tobii eye-tracker of the HTC-Vive VR headset is used for eye-tracking. A calibration is made after the tutorial session *regardless* of whether the current session would use gaze information. The purpose of eye-gaze is *not* disclosed to the subject.

3. **Briefing** Sophia would shuffle and then point out the location-pose pairing for the up-coming posing task session to the subject, before which it would be emphasized that the ordering of achieving the sub-goals could vary but the pairing must be correctly followed, as per the partially ordered task presentation in Figure 4($a$). Note that the same set of reference poses and locations, as illustrated in Figure 9($b$), is used for all sessions of all groups. The briefing phase is purposefully designed to be quick and vague, thereby creating the need of further guidance at different abstract levels later during the execution of the posing task. This video is a demo of the briefing phase.

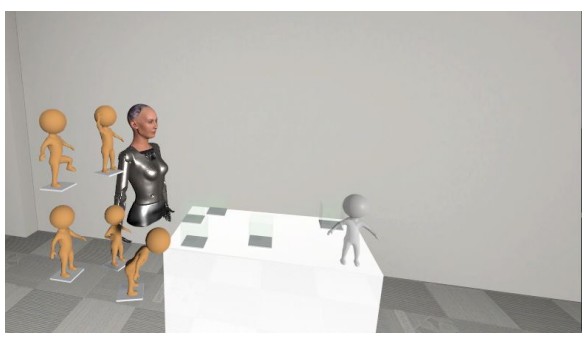 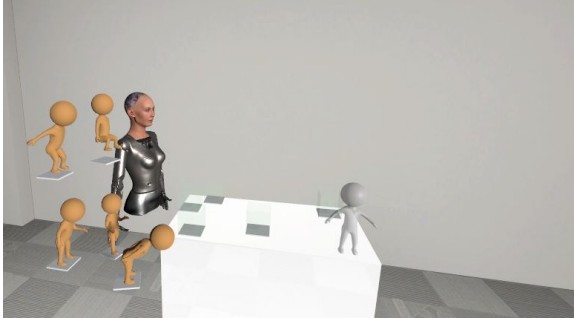

(*a*) Mock reference poses used during tutorial sessions.  (*b*) Reference poses used during experiment sessions.

Figure 9: Reference pose setup for tutorial sessions and experiment sessions.

4. **Task Execution** After the briefing the subject moves on to actually carry out the posing task with guidance from Sophia, controlled either by our framework or by the human wizard, with or without gaze. The entire task execution is recorded for replay later. Here is a demo of task execution.

5. **Evaluation** The subject was asked to make an assessment of the guidance Sophia provided after finishing the posing task. First the subject would be asked to watch the replay of the task execution process he / she just went through, during which process the subjects are required to try to recall his / her state and the progress of the task, based on which the subject would answer a set of questions regarding the timeliness and precision of guidance at the end of each cue. This assessment process is illustrated in this demo video and details of the questions that the subject would be prompted with are given in Appendix B.4. After watching the replay the subject would answer the SUS questionnaire detailed in Appendix B.5.

6. **Reward** The subject is rewarded 50 HKD per hour at the end of each experiment session. In total we spent 2650 HKD on subject compensation.

## B.4. Playback Questions

When watching the playback the subject is instructed to recall his / her state as well as the task progress. The subject would be prompted with the following questions at the end of every cuing action in the playback. The first question attempts to establish whether there should be guidance at that moment.

*Question 1* **Should there be a cue?**
*Question*: In retrospect, should there have been a cue at that moment?
*Instruction*: Answer this question by reflecting on your state and the progress of the posing task. Choose one of the options below by considering whether your were consciously seeking help at this moment and whether there is a mistake / an issue hindering task progress.
*Options*:

1. Yes. I wanted help.

2. Yes. I was making a mistake or having difficulty.

3. Yes. Both of the above are true.

4. No. Sophia should have remained silent.

5. None of the above. Please elaborate.

If the subject selects 4 in the first question, the evaluation for this cue terminates immediately. If the subject selects any other options, he / she would then be prompted with question 2 and 3 below.

### Question 2   Timing of the Cue

*Question*: Please evaluate the TIMING of the cue.
*Instruction*: Answer this question without considering the exact content of the cue. Instead choose from the options below by considering whether the timing of this cue is appropriate given the issue / your wish of getting help identified in Question 1. E.g., Maybe you have been looking for help for a long time, or maybe Sophia could wait till you finish what you have at hand before addressing an existing issue.
*Options*:

1. The cue should have come sooner.

2. The cue came at an appropriate time.

3. The cue should have come later.

4. None of the above. Please elaborate.

### Question 3   Content of the Cue

*Question*: Please evaluate the CONTENT of the cue.
*Instruction*: Answer this question by comparing the content of the cue with the issue present / the help sought after identified in Question 1 and choose from the options below. E.g., pointing out what joint to correct next might be too low-level when the issue is your forgetting the target pose, whereas pointing at the target pose is probably too generic to help you choose the correct arrow to use next.
*Options*:

1. The information given was too ABSTRACT or GENERIC.

2. The information was given at the correct level of detail.

3. The information given was too LOW LEVEL.

4. The information given was IRRELEVANT.

5. None of the above. Please elaborate.

### B.5. System Usability Scale

System Usability Scale (SUS) has been widely used in usability study (199, 1996)(Lewis, 2018). In our experiment we use SUS for an overall evaluation of the performance of Sophia serving as a helper to provide guidance. The instructions we give and the questionnaire answered by the subjects are shown below, whose results are normalized following the procedures described in (199, 1996).

*Instruction*: Please answer each item by marking a number to indicate how much you agree with each statement. Answer all items even if unsure of your answer. Note that "this system" stands for the Sophia robot serving as a helper in the task in the questions below. *Questions*:

1. I think that I would like to use this system frequently.

2. I found the system unnecessarily complex.

3. I thought the system was easy to use.

4. I think that I would need the support of a technical person to be able to use this system.

5. I found the various functions in this system were well integrated.

6. I thought there was too much inconsistency in this system.

7. I would imagine that most people would learn to use this system very quickly.

8. I found the system very cumbersome to use.

9. I felt very confident using the system.

10. I needed to learn a lot of things before I could get going with this system.

### B.6. Wizard-of-Oz Setup

Figure 10(*a*) gives an overview of the Wizard-of-Oz setup. The human wizard sits in a location hidden from the subject, monitoring the progress of the posing task on the screen and invoking cuing actions through a keyboard. The location-pose pairing and the difference between the puppet and the target pose could be seen on the monitor, as shown in Figure 10(*b*). The observations available to the wizard is restricted to be exactly the same as that when running our proposed framework. The demo video here illustrates the difference: for sessions without gaze the wizard could only see the arrow manipulation made by the subject, whereas for session with gaze the wizard sees both the gaze point and the arrow manipulation. In either case, no other behavioral information from the subject is available to the wizard.

To offer guidance to the subject, the wizard is trained to remember the correspondence between key-strokes and cuing actions – a list is also available for reference during the experiment. The wizard could either choose the level of abstraction of the cue to offer and reuse our proposed algorithm's optimal branch to choose the exact task which Sophia would

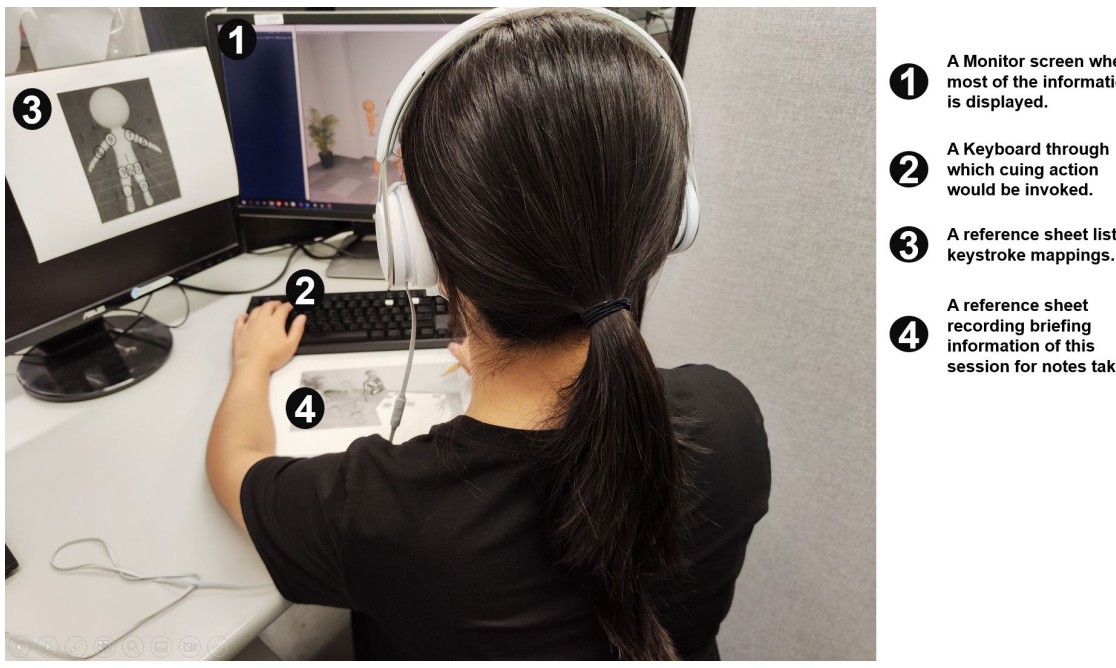

*(a)* Overview of the workspace of the human wizard.

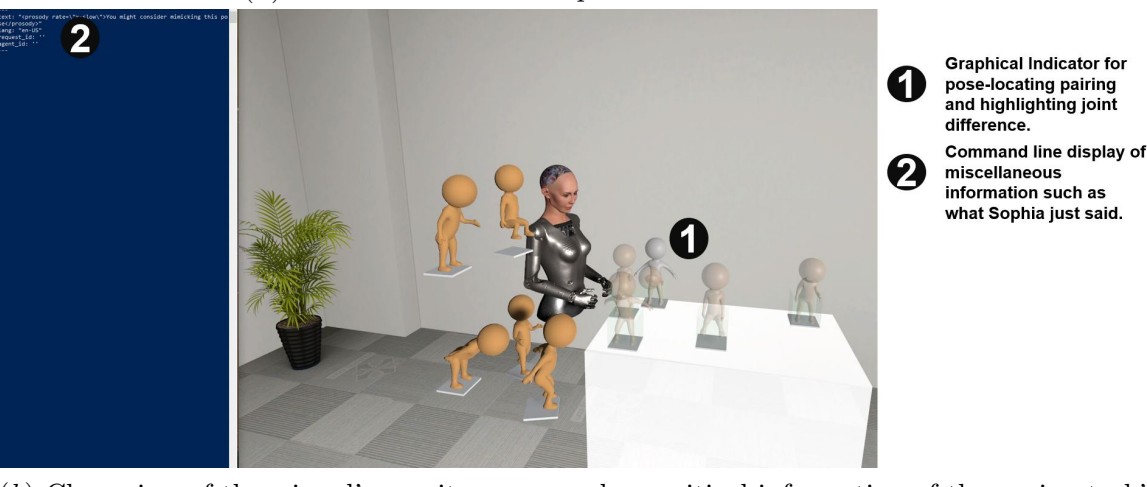

*(b)* Close view of the wizard's monitor screen where critical information of the posing task's execution is displayed.

Figure 10: Illustration of the human wizard's work setting.

cue, or manually specify exactly which task should Sophia talk about. Although this is certainly not the most ideal interface to be used by a human wizard, later remarks from the wizard suggest that the interface was certainly not a major constraint.

During training the wizard tried the posing task repeatedly, taking the role of the subject as well as the role of the wizard. In the former case the wizard also completed the evaluation by watching the replay and answering the questions (Appendix B.4,Appendix B.5). Through this process the wizard became familiarized with the evaluation criteria of the posing task.

The wizard is paid by a base amount of 50 HKD per hour for both training and experiment sessions. To further create incentives, the wizard was informed that the base reward would be scaled according to the quality of guidance assessed by subjects from the Wizard Group. Specifically, if the average ratio of timely and precise cues of the Wizard Group sessions with gaze is $x$ percent higher then that of the Automated Group, the wizard's reward for those sessions (again with gaze) would be multiplied by $1 + 0.1x$. Same reward scheme is adopted for sessions without gaze except that the multiplier is changed to $1 + 0.01x$. But the exact multipliers were not disclosed to the wizard until Wizard Group experiments have been completed. In total we paid the wizard by a amount of 2060 HKD. Note that during wizard-group sessions the wizard was *not* present when the subjects performed the evaluation. Instead the wizard was only informed about the subjects' opinions and granted access to the replays *after* finishing *all* wizard group session.

