# OpenReview forum: "Intention Estimation via Gaze for Robot Guidance in Hierarchical Tasks"
_NeurIPS.cc/2022/Workshop/GMML — Gaze Meets ML 2022 Oral_

### Official Review · Reviewer_AHpX · 2022-10-11
**A good paper showing how the addition of intention gaze aids in robot guidance for task completion**

**Rating:** 7
**Confidence:** 4

**Review:**

## Quality
The experimental design was sound. The only additional metrics I would have liked to have seen are the time to completion of the task and the mean squared error in the joint actuation path. I think this would have been more objective and informative than asking the subjects to verbally rank the timeliness and precision of the guidance. Nevertheless, I don't think these measures would have changed the authors' conclusions. Perhaps there could even be a random delay in the eye tracking data to give a more granular measure of the importance to the gaze feedback.

I applaud the authors for such a detailed explanation of the limitations of their assumptions. I agree that the assumption of independence between the motion and intention of the non-leaf tasks is too much of a simplification. My guess is that if the task hierarchy were more complex (more levels, different lengths in branches) this assumption would significantly change the results. The authors correctly acknowledge this flaw.

##  Clarity
Excellent clarity. I thought the analogy of the apprentice cook and the master chef making one of two recipes was useful. This analogy helped crystalize the experiment in my mind. The figures were clear and conveyed the results well. Although I could not view the supplemental video I'm sure it would have been useful to the reader.

## Originality
The authors state "... to the best of our knowledge eye gaze has not been exploited for robot guidance". I don't doubt the authors' statement, but want to point out some close papers that should be reviewed and referenced:
* Li, Yin, Miao Liu, and James M. Rehg. "In the eye of beholder: Joint learning of gaze and actions in first person video." Proceedings of the European conference on computer vision (ECCV). 2018. - The authors add gaze estimation to augment a task prediction model.
* Huang, Sandy H., et al. "Nonverbal robot feedback for human teachers." arXiv preprint arXiv:1911.02320 (2019). The authors are essentially doing the reverse task: namely, the robotic teacher is providing its model gaze prediction as a cue to the human learner to help with the task recognition and completion.
* Emhardt, SN, van Wermeskerken, M, Scheiter, K, van Gog, T. Inferring task performance and confidence from displays of eye movements. Appl Cognit Psychol. 2020; 34: 1430– 1443. https://doi.org/10.1002/acp.3721. The authors investigate to what extent observers can make inferences from other people's eye-movement displays. Of course, this does not include arm movement, but it does show that task inference for the teacher is significantly improved when shown the eye-tracking data of the student.

## Significance of the work
The work emphasizes that the inclusion of eye movement data must be included in a hierarchical task model. This should be important in guiding future designs of task modelling.

## Additional questions
1. In section A.2 "Intention Estimation" (line 507): Did the authors measure any change in the frequency of type 0 "looking at irrelevant entities" when no help was needed? If there is a change, then it might be useful to trigger the task model to provide help.

### Pros:
* Well written, very clear
* Good use of relevant analogies
* Thorough experimental design
* Excellent discussion of the limitations in the study

### Cons:
* Line 205: The word "manipulate" should be "manipulated".
* Would like authors to consider the additional references if they are relevant.

---

### Official Review · Reviewer_Qntx · 2022-10-19
**Novel and well written paper that demonstrates how gaze can improve cues in human agents performing hierarchical tasks**

**Rating:** 9
**Confidence:** 3

**Review:**

Page 2, figure 1. Very clear example that provides the reader with the motivation and the notion of hierarchical tasks.

There may be a limitation given the strict ordering of tasks in a plan. For example, how does the system perform in when the human has the freedom to perform tasks out of order? For example, the cook could start thawing the meat prior to peeling the onion. Would that compromise the robustness of the system?

Another consideration is optionality of the tasks. Onions may be optional in a recipe, or I may choose to skip it entirely due to my preferences, while still achieving my goal of cooking recipe 1. How does the system handle optionality, or human agent creativity when subtasks are handled differently?

In section 4.1, line 199, it is unclear why the authors chose to combine puppet translation to a location and controlling the puppet to mimic a pose as "pose mimicry" task? As the video demonstrates the cues will be distinct for those tasks.

in the experiment, was there a mechanism to evaluate if the agent needs guidance v/s wants guidance, as described in section 3.3?

What happens when the cue is missed, or misunderstood? How are the probabilities updated?
Section 4.2, consider moving some of the content is appendix b.6 into the section. The section cannot be clearly understood independently without referring to the appendix. At a minimum provide 1 line description of the setup for each of the following: Automation Gaze, Automation no-gaze, Wizard-Gaze, Wizard no-gaze

The discussion section nicely describes the limitations, I would have liked to see more discussions on potential failure scenarios that would set the stage for future work.

---

### Meta-Review · Area_Chair_tJZ7 · 2022-10-20

**Recommendation:** Accept (Oral)
**Confidence:** 4

**Metareview:**

This work investigates how eye gaze observations can improve hierarchical robotic task guidance. Reviewers have positively commented on the contributions of this work, as well as the detailed explanations of the motivation, scope, and limitations. Reviews have also listed several additional references and raised important questions, some of which the AC hopes can be addressed in the camera-ready version. Overall, this is well-presented work of great interest to the target audience; recommending acceptance.

---

### Decision · Program_Chairs · 2022-10-20

Accept (Oral)